# Optimization of On-Street Parking Charges Based on Price Elasticity of the Expected Perceived Parking Cost

**Jun Li \*, Sifan Wu and Xiaoman Feng**

School of Intelligent Systems Engineering, Sun Yat-sen University, Guangzhou 510000, China;
wusf@mail2.sysu.edu.cn (S.W.); f18256192277@126.com (X.F.)
\* Correspondence: stslijun@mail.sysu.edu.cn

**Abstract:** Price discrimination is widely employed to regulate on-street parking behaviors to provide better service to users, and the prices are usually set according to the occupancy of parking spaces without direct consideration of user perception. A binary logit-style choice model is built to describe the parking choice between on-street parking and off-street parking. A new index, named the price elasticity of expected perceived parking cost, is proposed to evaluate users' response to parking charge. Based on the theory of second-degree price discrimination, three user types are defined according to the parking duration, namely, the preferred users, the neutral users, and the non-preferred users. The optimized parking prices are calculated by the proposed index. A case study of Guangzhou's on-street parking is presented. It is found that the current pricing scheme for Type-I Zones (High Demand Zones) is reasonable, while the pricing scheme for the Type-II Zones (Low Demand Zones) does not achieve the objectives of usage optimization of on-street parking spaces. An optimized price scheme for the Type-II Zones is proposed to achieve the usage optimization of on-street parking spaces for short-term parking.

**Keywords:** on-street parking; expected perceived cost; price elasticity; second-degree price discrimination



## 1. Introduction

On-street parking spaces are located on the roadside and the number of parking spaces is limited, therefore they are suitable for short-term temporary parking users and should have a higher parking turnover rate to provide service to more users [1,2]. On the contrary, long-term parking is suitable for off-street parking. Long-term on-street parking reduces on-street parking turnover rate, meanwhile, long-term off-street parking can improve parking safety and reduce the risk of vehicle damage [3]. The parking fee is the most critical consideration factor when travelers choose the parking location, and the travelers always choose the one that maximizes their utility, so that the parking charges have great impacts on the users' parking behavior [4–6]. Low on-street parking charges lead to high parking occupancy rates, long parking durations, and inefficiency of usage of off-street parking resources. When on-street parking spaces are occupied by people who park for a long time, the parking turnover rate is low, causing a large number of vehicles to cruise for finding parking spaces [7,8] and even traffic congestion [9,10]. Studies have found that in urban centers, especially the central business district (CBD), the increase of pollutant level is mainly related to on-street parking [11,12]. Some scholars have observed the influences of various travel factors on travelers' behavior are based on the heterogeneity of travelers, analyzed the utility function of various parking methods, and optimized the price with the ideal parking sharing rate [13–16]. Others have used marginal cost-benefit analysis to take factors of environmental protection and living standards of surrounding residents [17], set charging threshold [18], and set pricing into consideration in order to maximize social benefits [19]. In recent years, some scholars have combined expected utility and prospect theory with travel mode selection, and used the combination of travel mode sharing logit model and private car traveler satisfaction function to build a price-setting model [20,21].

More and more scholars have realized that considering the sensitivity of parking charges for the different types of users is essential for price setting [2]. The price discrimination mechanism is proposed from the perspective of users so that the charging prices within the price range are acceptable to users in order to maximize the influence and guidance of the charging mechanism on consumer behavior [22].

The ratios of on-street parking are usually used as the indexes to determine the prices while creating the pricing strategies. However, those methods have failed to distinguish the different types of users [9], which cause the failure of meeting the main objective of providing the on-street parking space, namely, providing service for the short-term parking users. According to the previous researches, parking time, walking distance and satisfaction degree of parking behavior of travelers have significant guidance for the formulation of parking price [23], and it is argued that the pricing should be charged according to the sensitivity differences in various parking time. In order to achieve the purpose of providing parking spaces to short-term parking, many cities have adopted a progressive charging mechanism: the parking time is divided into several stages with considerable price rising for longer parking period. However, such a mechanism is lack of systematic theoretical support for the price standards for each stage and does not include the impact of other alternative parking methods. Progressive pricing is an application of price discrimination, and nowadays has become a popular method to establish charging mechanisms in numbers of cities [24,25] since the price discrimination can distinguish users and guide users' behavior. The functions of parking price on parking behavior can be enhanced more accurately by imposing different charging rates on the users with different parking periods in different areas [26,27], but there is a need for the quantitative analysis of price discrimination.

The reasonable parking charges are imperative to achieve the goal of optimizing on-street parking. Charging parking fees can filter out some unnecessary parking demands and save the parking spaces for users who are willing to pay more for parking [28]. Therefore, on the basis of considering users' willingness to pay, many scholars have introduced the concept of elasticity as a pricing mechanism measurement index [29]. Price elasticity refers to the elasticity of demand pricing and the sensitivity of the corresponding change in the demand for a certain product when the price of the product changes [30,31]. In the field of transportation, the price elasticity is often used as an index to evaluate the pricing mechanism. For example, some scholars analyzed travel and parking behaviors based on price elasticity theory and found that travelers are more willing to switch parking locations than to change travel modes [32]. Users who often park in a certain area are more sensitive to the price of the parking area than those who park in this area occasionally, whose price elasticity is higher [33]. It is found that the longer the parking time of users, the more sensitive they are to the price changes, and the expected price as well as the price elasticity must be taken into account in order to charge users with different parking times.

The aim of this study is to develop a method to optimize the on-street parking price mechanics to give better experiences for the users and achieve the goals of the on-street parking, namely, providing parking services for temporary parking. By adjusting the parking distribution among multiple parking lots, the number of cruising vehicles on the road can be reduced [34]; furthermore, traffic jams and environmental pollution can be decreased [35], which is conducive to the sustainable development of the city. The concept of the expected perceived parking cost is proposed to represent the expected perceived utilities of the satisfaction function—to describe the situation that the users have multiple parking alternatives. The utilities of various parking alternatives are assumed to be the functions of parking costs, which provide a pricing basis for the establishment of progressive charging mechanism of on-street parking. The users are divided into multiple groups according to the objects of parking administrator, and the price elasticity of the expected perceived parking cost is proposed to optimize the parking prices for each defined user group.

## 2. Methodology

The progressive charging mechanism for the on-street parking is widely implemented to restrict the undesirable usage of on-street parking, especially long-term parking. The off-street parking is selected as the reference mode to the on-street parking in this study. It is assumed that the administrator wishes to promote the short-term parking while reducing the long-term parking by the price discrimination. The users are divided into three groups according to the parking durations, namely, the preferred users, the neutral users, and the non-preferred users. The binary logit-style choice model will be proposed to simulate the parking choice behaviors between on-street and off-street parking. The value of price elasticity of expected perceived parking cost is suggested for each group to achieve the goals of the price discrimination with direct consideration of user senses to parking prices.

### 2.1. Parking Utility Function

It is assumed that the users have only two choices of on-street parking and off-street parking. The utility of each alternative is assumed to be the linear function of cost. Noticing that the logit model is only determined by the differences among the alternatives [36], the constant value of the utility function of on-street parking is set to zero without loss of generality. Since the off-street parking requires more time to find a parking space and then walking to the final destination than on-street parking [37], the extra cost needed for off-street parking is defined as the parking operation time, which is set to a constant for simplification purposes, no matter how long the parking duration is. The utility function of the two parking modes can be expressed as in Equation (1):

$$U_i = -\theta C_i \quad (i = c, g) \tag{1}$$

where $U_i$ is the utility of mode $i$; $C_i$ is the cost of mode $i$; $c$ and $g$ represent on-street parking and off-street parking, respectively.

$$U_c = -\theta v \tag{2}$$

$$U_g = -\theta(v_g + \omega) \tag{3}$$

where $v$ is the parking charge of on-street parking; $v_g$ is the parking charge of off-street parking; constant value $\omega$ represents the time cost of off-street parking operation time; $\theta > 0$ is the parameter. It should be noticed that each user group gets its own parameter $\theta$, and there would be $n$ parameters $\theta_1, \theta_2 \ldots \theta_n$ when there are $n$ user groups.

### 2.2. Expected Perceived Parking Cost

Considering the situation that the users are facing multiple parking choices, the users have an overall value perception for the parking choices. The more options available and the lower the cost of each mode, the better users will feel. The overall perceived value of the users can be calculated by the satisfaction function, defined as follows [38]:

$$\widetilde{S} = E\left[\max_i\{U_i\}\right] = E\left[\max_i\{-\theta C_i\}\right] = \ln \sum_i^{c,g} \exp(U_i) \tag{4}$$

The satisfaction function is the overall utility rather than the monetary value, which is difficult to evaluate the user experience directly. In this study, the expected perceived parking cost (EPPC) is proposed to represent the perceived value of users, transforming the utility research into a more intuitive price, which is defined as follows according to Equation (1):

$$\widetilde{C} = E\left[\min_i\left\{-\frac{U_i}{\theta}\right\}\right] = -\frac{\widetilde{S}}{\theta} \tag{5}$$

It is worthwhile to note that the derivative of $\widetilde{C}$ with respect to $C_i$ is the probability the mode $i$ is chosen, shown as follows:

$$P_i = \frac{\mathrm{d}\widetilde{C}}{\mathrm{d}C_i} \tag{6}$$

It is assumed that the cost of off-street parking is constant in this study, so that the effect of on-street parking price on the EPPC can be more straightly analyzed. The relationship between EPPC and the on-street parking cost is shown in Figure 1.

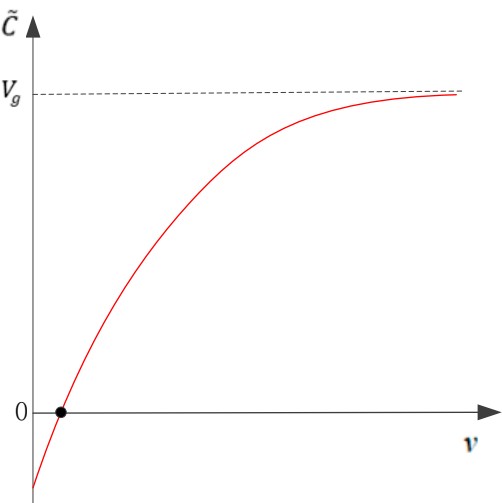

**Figure 1.** EPPC diagram.

As shown in Figure 1, the EPPC increases with the increase of the actual charging price, but the EPPC is always less than the cost of any alternative. It is possible that the EPPC is negative when the parking price is small enough. With the increase of the parking price, the EPPC changes from negative to zero. With the increase of the parking cost, the EPPC increases at a decreasing speed and then tends to be flat, infinitely approaching to the cost of reference parking mode $V_g$.

### 2.3. Optimization of Price Discrimination

The price elasticity is widely used to set the price for the given service since it can effectively represent the effects of price change. In this study, the price elasticity of the EPPC with respect to the on-street parking price can be expressed as Equation (7):

$$e(v) = \frac{d\widetilde{C}}{dv} \Big/ \frac{\widetilde{C}}{v} = -P(v)\frac{v}{\widetilde{C}} \tag{7}$$

where $e(v)$ is the price elasticity of EPPC to the on-street parking price; $P(v)$ is the probability of choosing on-street parking when the on-street parking price is $v$. The diagram of the price elasticity of EPPC is shown in Figure 2.

Since $\widetilde{C}$ can be zero or even negative, there exists a price $V_0$ where the elasticity of EPPC tends to be infinite and the price has the greatest influence on the EPPC. In real-world applications, the prices of on-street parking are usually high enough so that $\widetilde{C}$ is positive.

As shown in Figure 2, it is easy to set the desired price according to the price elasticity. For example, if the administrators wish that the users have moderate impact of price change and set the price elasticity falling the interval $(E_1, E_2)$, the reasonable price should be between $V_1$ and $V_2$.

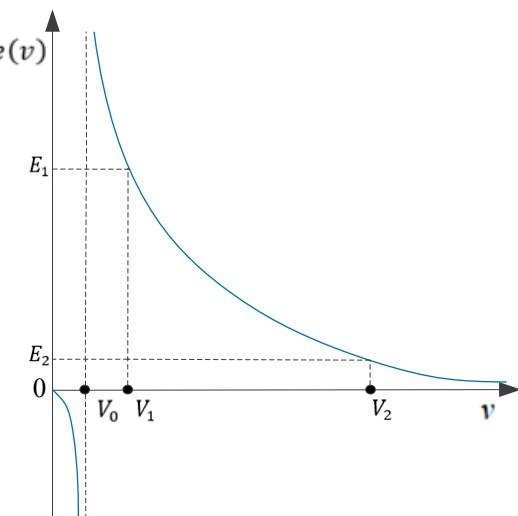

**Figure 2.** Price elasticity of EPPC diagram.

Now, considering the progressive charging mechanism for the on-street parking, it is important to encourage the users to take short-term on-street parking and reduce the long-term on-street parking. Three types of users are defined in this study according to the parking charging stages, namely, preferred users, neutral users and non-preferred users. Of course, further refined groups can be defined as required.

The preferred users are defined to be the users who park for a short time and are preferred by the administrator, and the parking charge rate is referred as the first stage. As shown in Figure 2, setting price in the interval $(0, V_1)$ has a great impact on the perceived value of users' parking behavior, when the users whose price elasticity is greater than $E_1$ and the on-street parking price is less than $V_1$. The EPPC change is approximate or greater than the change of the on-street parking price, which benefits the users. In the practice, the approximation can be achieved by set $E_1$ to $0.8 \sim 1.0$.

The neutral users are those who park for a relatively longer parking time. The price elasticity falls into interval $(E_1, E_2)$ and the price should be set between $V_1$ and $V_2$, as shown in Figure 2. In such a scenario, the price setting should ensure that some users will give up on-street parking because their EPPC is lower than the actual on-street parking price. At the same time, the elasticity should be greater than a small number so that users will not completely move off, which ensures no waste of on-street parking resources. Usually setting $E_2$ to $0.10 \sim 0.20$ should be an appropriate value for the lower limit. The greater the price elasticity of EPPC, the less likely users are to change their parking methods.

The users who park for long time are defined as non-preferred users, since the off-street parking should be a more appropriate choice and be desired by the administrator. In order to restrict the long-term on-street parking, the price should be raised to the interval with a small price elasticity of EPPC. As the on-street parking price continues to increase, the elasticity of the EPPC decreases to a flat section and the gap between the perceived price of expectation and the actual price gradually becomes larger. Continuing to increase on-street parking prices will not affect the overall perceived value of users, and the EPPC of users will not change. In this case, the price reaches the bottom line of effective pricing, and pricing in this range can make non-preferred users give up on-street parking.

The optimized price elasticity of EPPC at each stage is shown in Table 1. The price elasticity of EPPC can not only be used to obtain the lowest and highest prices in line with consumer expectations, but also be used to verify whether the existing pricing scheme is reasonable.

**Table 1.** Optimized price elasticity of EPPC at each stage.

| User Type | Parking Stage | Price Elasticity of EPPC |
|---|---|---|
| Preferred users | I | $e < 0$ or $e \geq E_1$ |
| Neutral users | II | $E_2 \leq e < E_1$ |
| Non-preferred users | III | $e < E_2$ |

## 3. Case Study

Guangzhou City is selected as our case study. A new parking charge policy has been implemented in Guangzhou since 2020, which stipulates that the on-street parking in the Type-I Zones and the Type-II Zones should be charged by different amounts, and the progressive charging method is adopted to divide the parking time into three stages. According to the parking fee standard for temporary parking spaces on roads in Guangzhou, parking fees are charged from 7:30–21:30 on weekdays and 10:00–21:30 on non-working days. Parking is free of charge except for the charging period. The charging policy stipulates that there is no charging for vehicles parked for less than 15 min, and the half hour charging scheme (RMB/30 min) is adopted. The charging mechanism is shown in Table 2.

**Table 2.** Charge standard of on-street parking in Guangzhou.

| Area | Parking Type | Parking Stage/Parking Duration (RMB/30 min) | | | Ceiling Price (RMB) |
|---|---|---|---|---|---|
| | | I/Within 1 h | II/1 h~3 h | III/Over 3 h | |
| Type I | On-street | 5 | 8 | 13 | 328 |
| | Off-street | | 4 | | 128 |
| Type II | On-street | 2 | 3 | 5 | 26 |
| | Off-street | | 2.5 | | 30~60 |

The Type-I Zones refer to the high demand areas, such as hospital, government, resources trading center, shopping malls, business district, etc., and the Type-II Zones refer to the low demand areas, that is, all other areas excluding the Type-I Zones. There are two principles in the pricing of urban public parking spaces:

1. The parking price in Type-I Zones should be higher than that in the Type-II Zones;
2. The on-street parking price should be higher than the off-street parking price.

### 3.1. Data

The Renmin North Road of Yuexiu District (the Type-I Zones), Huandao Road and Taigucang Road of Haizhu District (the Type-II Zones) are selected for experimental study; the average parking operation time of off-street parking is assumed as 10 min, and the value of operation time can be calculated by Equation (8):

$$\omega = \mu t \tag{8}$$

where $\omega$ is the time cost of off-street parking operation time; $\mu$ is the unit time value, and $t$ is the off-street parking operation time. According to the previous studies, the value of operation time can be calculated from the average wage of the city [39]. In this study, the value of operation time is set to 1 RMB/min, so that the total value of operation time of off-street parking is 10 RMB.

The binary logit-style choice model is used to express the probability of on-street parking being selected when the parking fee is $v$. The probability of one parking mode is determined by the utility of two parking modes, where one is chosen because of its greater utility than the other [36], which is shown in Equation (9):

$$P(v) = \frac{exp(U_c)}{exp(U_c) + exp(U_g)} \tag{9}$$

Online questionnaires were used to investigate the changes in the proportion of on-street parking and off-street parking when users have different parking durations in the Type-I Zones and the Type-II Zones during the implementation of the charging mechanism as shown in Table 2. A total number of 400 questionnaires were sent out and 384 valid data were collected. The probabilities of on-street parking are chosen when the parking durations are half an hour, 1 h, 2 h, 3 h, 4 h, and 5 h, and the parameter $\theta$ is obtained from Equation (6). The data and $\theta$ are shown in Table 3.

**Table 3.** Parking proportion and parameter of on-street parking.

| Parking Stage | Parking Duration | Type-I Zones | | Type-II Zones | |
|---|---|---|---|---|---|
| | | *P(v)* | $\theta$ | *P(v)* | $\theta$ |
| I | 30 min | 0.90 | 0.244 | 0.99 | 0.354 |
| | 1 h | 0.85 | 0.217 | 0.97 | 0.200 |
| II | 2 h | 0.46 | 0.160 | 0.63 | 0.133 |
| | 3 h | 0.23 | 0.151 | 0.46 | 0.160 |
| III | 4 h | 0.09 | 0.089 | 0.64 | 0.144 |
| | 5 h | 0.05 | 0.067 | 0.82 | 0.169 |

### 3.2. Parking Charge in Type-I Zones

Based on the data in Table 3, the parameters and price elasticity of EPPC in the Type-I Zones are obtained from Equations (1)–(9), as is shown in Table 4.

**Table 4.** Analysis of parking charge in Type-I Zones.

| Parking Stage | Parking Duration | On-Street (RMB) | Off-Street (RMB) | Parameter $\theta$ | $\tilde{C}$ | *e(v)* |
|---|---|---|---|---|---|---|
| I | 30 min | 5 | 4 | 0.244 | 4.568 | 0.985 |
| | 1 h | 10 | 8 | 0.217 | 9.251 | 0.919 |
| II | 2 h | 26 | 16 | 0.160 | 21.162 | 0.565 |
| | 3 h | 42 | 24 | 0.151 | 32.271 | 0.299 |
| III | 4 h | 68 | 32 | 0.089 | 40.941 | 0.149 |
| | 5 h | 94 | 40 | 0.067 | 49.233 | 0.096 |

According to Table 4, the elasticity of users' EPPC to the existing parking charges in the Type-I Zones of Guangzhou can be drawn as Figure 3.

By observing the image of the EPPC in the Type-I Zones and comparing the reasonable range of the price elasticity of EPPC of users with different parking durations in Table 1, it can be judged whether the pricing is reasonable. The analysis results are as follows:

1. The users in the first stage are the preferred users, and the elasticity of users' EPPC to on-street parking charges is above 0.9. The expected charging price of users is close to the actual charging price. Short-term parking users will choose on-street parking and the pricing is reasonable.

2. In the second stage, users are neutral users, and the elasticity of users' EPPC to on-street parking charges is in the range of 0.15~0.9. As the parking time becomes longer and the price difference between the two parking modes becomes larger, more users tend to give up on-street parking and shift to off-street parking. Therefore, the pricing in the second stage is reasonable.

3. Users in the third stage are non-preferred users, and the price elasticity of EPPC is less than 0.15. The change of on-street parking charge has less and less impact on the change of EPPC. After realizing the price difference between the two parking modes, users would choose off-street parking rather than on-street parking. Therefore, the price of on-street parking in the Type-I Zones is reasonable.

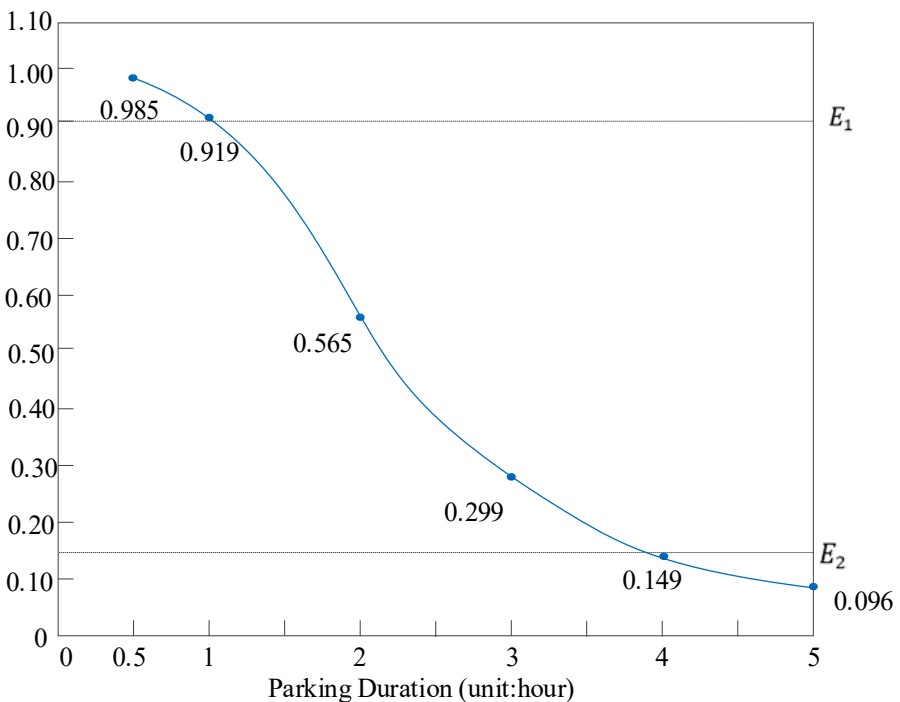

**Figure 3.** Price elasticity of EPPC in the Type-I Zones.

### 3.3. Parking Charge in Type-II Zones

Based on the data in Table 3, the parameters and price elasticity of EPPC in the Type-II Zones are obtained from Equations (1)–(9), as is shown in Table 5.

**Table 5.** Analysis of parking charge in Type-II Zones.

| Parking Stage | Parking Duration | On-Street (RMB) | Off-Street (RMB) | Parameter $\theta$ | $\tilde{C}$ | $e(v)$ |
|---|---|---|---|---|---|---|
| I | 30 min | 2 | 5 | 0.354 | 1.972 | 1.002 |
| | 1 h | 4 | 5 | 0.200 | 3.904 | 0.993 |
| II | 2 h | 10 | 10 | 0.133 | 12.532 | 0.805 |
| | 3 h | 16 | 15 | 0.160 | 21.161 | 0.561 |
| III | 4 h | 26 | 20 | 0.144 | 22.901 | 0.727 |
| | 5 h | 26 | 25 | 0.169 | 24.823 | 0.859 |

According to Table 5, the elasticity of users' EPPC to the existing parking charges in the Type-II Zones of Guangzhou can be drawn as Figure 4.

By observing the EPPC in the Type-II Zones and comparing the reasonable range of the elasticity of users' EPPC with different parking durations in Table 1, it can be judged whether the pricing is reasonable. The analysis results are as follows:

1.  The users in the first stage are the preferred users, and the elasticity of users' EPPC to on-street parking charges is above 0.9. The price of on-street parking in the Type-II Zones is reasonable.

2.  The elasticity of users' EPPC to on-street parking charges ranges from 0.15 to 0.9. Although it is within a reasonable range, the elasticity is larger than the one of Type-I Zones, exceeding 0.5 and leading only a few users to give up on-street parking. Therefore, it is necessary to appropriately raise the pricing and reduce the elasticity to make more users shift to off-street parking in the second stage.

3.  Users whose parking time is more than 3 h in the third stage are non-preferred users and should shift to off-street parking. The ceiling price of 26 RMB is lower than the off-street parking price, which violates the pricing principle of on-street parking. The

price elasticity of EPPC is high, and it appears that most users who park for more than three hours would choose on-street parking. Therefore, the setting of charging and ceiling price in the third stage is unreasonable and should be improved.

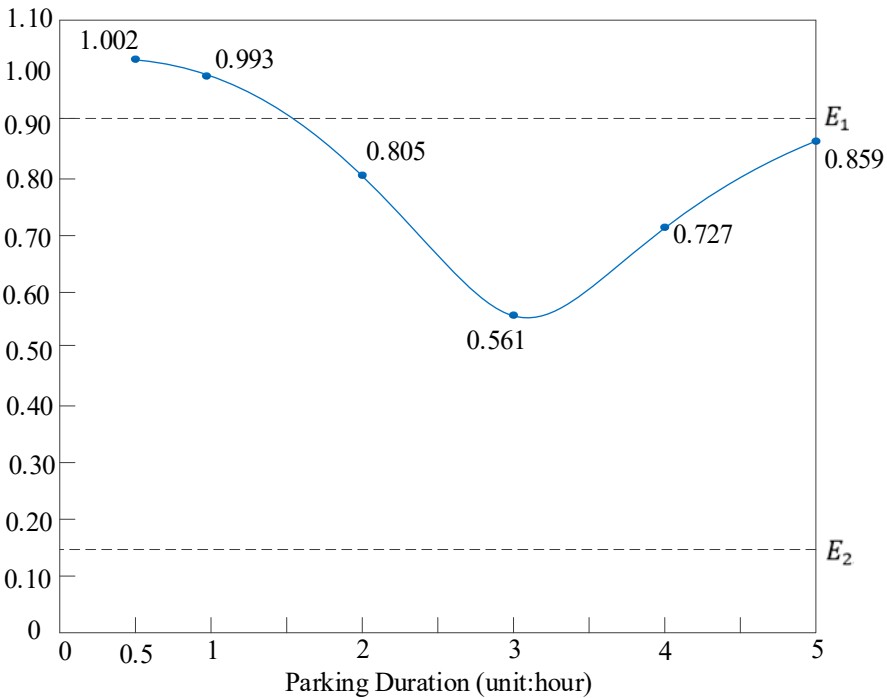

**Figure 4.** Price elasticity of EPPC in Type-II Zones.

*3.4. Price Optimization*

The price elasticity of the EPPC of the progressive charging policy in the Type-I Zones of Guangzhou City conforms to a reasonable elastic range, which allows on-street parking to serve short-term parking users while off-street parking serves long-term parking users. Although the elasticity of the second stage is within a reasonable range, the price elasticity of EPPC of three-hour parking is above 0.5 and most of the neutral users choose on-street parking. Moreover, from the fourth hour on, the price elasticity of EPPC in the third stage becomes larger, which not only fails to inhibit the long-term on-street parking behavior, but also shows that the longer the parking time is, the more users choose on-street parking.

The parameter $\theta$ of 3 h and 4 h parking is taken into the second and third stages, and the extra off-street parking cost is taken into Equations (5) and (7). With the on-street parking price charging within 3 h and 4 h as the independent variable, it reflects that the parking price in the Type-II Zones should be lower than that in the Type-I Zones (parking in the Type-I Zones costs 42 RMB for 3 h, 68 RMB for 4 h) according to the policy principle; long-term on-street parking price charging should be more than the one of off-street parking (off-street parking in the Type-II Zones costs 25 RMB for 3 h, and 30 RMB for 4 h). The change of EPPC and price elasticity of EPPC are shown in Figure 5.

As shown in Figure 5, the on-street parking price with an elasticity of 0.5 for 3 h parking period in Type-II Zones is 27.47 RMB, and the on-street parking price with an elasticity of 0.15 for 4 h parking period is 45.52 RMB, which is the minimum price meeting the elasticity requirements. For the convenience of charging in half an hour, the bottom price of parking is set for 3 h as 28 RMB and for 4 h as 46 RMB. Therefore, the price mechanism in the first stage of the optimized charging mechanism remains unchanged. In the second stage, the price charging is 6 RMB/30 min. In the third stage, the price charging is 9 RMB/30 min for on-street parking. According to the ceiling price setting of off-street parking in the Type-II Zones, the ceiling price of on-street parking in the Type-I Zones is the same as the price of on-street parking for 8 h. The optimized pricing mechanism analysis of the Type-II Zones is as shown in Table 6.

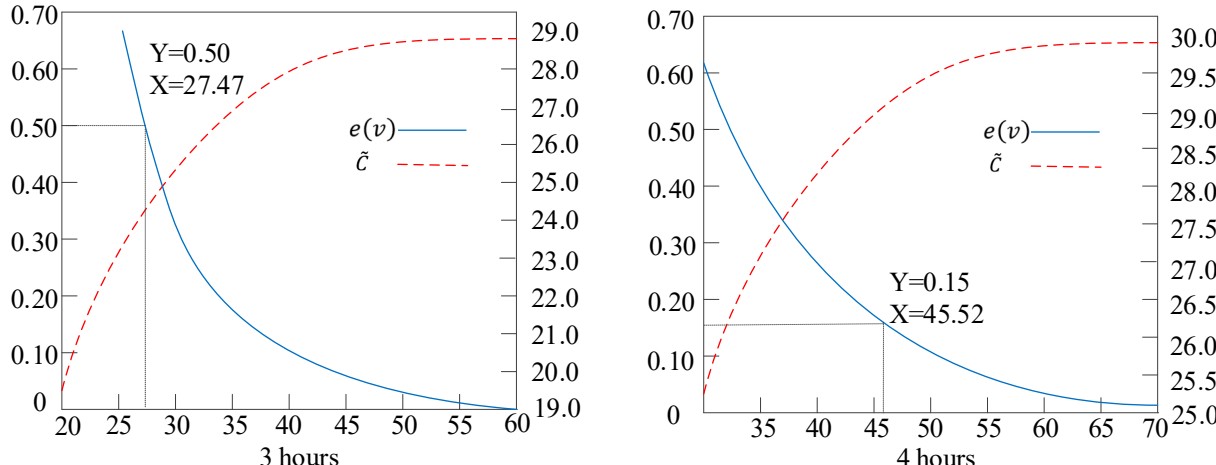

**Figure 5.** Price elasticity of EPPC changes with parking price in the Type-II Zones for 3 h and 4 h.

**Table 6.** The optimized pricing mechanism analysis.

| Parking Stage | Parking Duration | On-Street (RMB) | Off-Street (RMB) | $\tilde{C}$ | $e(v)$ |
|---|---|---|---|---|---|
| I | 30 min | 2 | 5 | 1.972 | 1.002 |
| | 1 h | 4 | 5 | 3.904 | 0.993 |
| II | 2 h | 16 | 10 | 12.532 | 0.802 |
| | 3 h | 28 | 15 | 21.191 | 0.500 |
| III | 4 h | 46 | 20 | 29.341 | 0.150 |
| | 5 h | 64 | 25 | 34.963 | 0.014 |

In this pricing scheme, the elasticity of parking EPPC of on-street parking in Type-I Zones and the Type-II Zones of Guangzhou changes with the increase of parking time, as is shown in Figure 6.

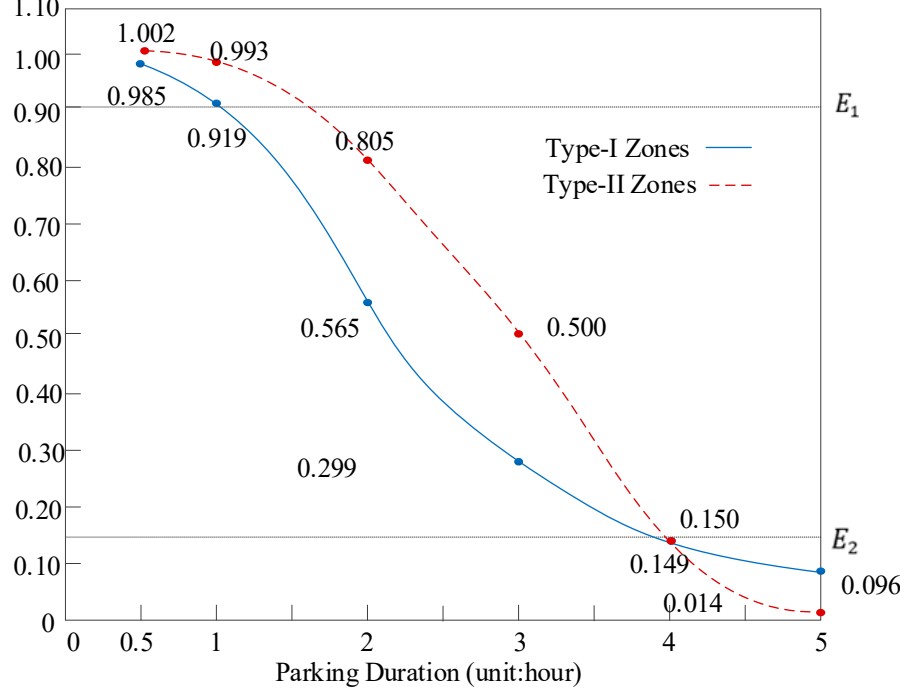

**Figure 6.** Optimized price elasticity of EPPC.

As shown in Figure 6, the optimized pricing mechanism can maintain the price elasticity of EPPC within a reasonable range and meet the policy principle. The specific pricing scheme is as shown in Table 7.

**Table 7.** Optimized charge standard of on-street parking in Guangzhou.

| Area | Parking Type | Parking Stage/Parking Duration (RMB/30 min) | | | Ceiling Price (RMB) |
|------|------|------|------|------|------|
| | | I/Within 1 h | II/1 h~3 h | III/Over 3 h | |
| Type I | On-street | 5 | 8 | 13 | 328 |
| | Off-street | | 4 | | 128 |
| Type II | On-street | 2 | 6 | 9 | 118 |
| | Off-street | | 2.5 | | 30~60 |

## 4. Conclusions

A method to optimize the pricing mechanism for on-street parking is proposed to provide proper on-street parking spaces to the users who need them the most and park for a short-term period, which is desired by the administrator. A binary logit-style choice model is proposed to simulate the user choice behaviors when they face the options of on-street and off-street parking. The concept of expected perceived parking cost is introduced for the quantitative analysis of user experience. It is suggested that the price elasticity of expected perceived parking cost with respect to the on-street parking price is a suitable index to optimize the price setting. Three user groups, namely, preferred users, neutral users and non-preferred users are proposed for the price discrimination. The price elasticity can be set to greater than $E_1$, between $E_1$ and $E_2$, and less than $E_2$ for preferred users, neutral users and non-preferred users, respectively, while $E_1$ can be set to 0.8~0.9 indicating that the parking price change has almost the same monetary impact on the welfare of the users, and $E_2$ can be set to 0.1~0.2 indicating that the parking price change has little monetary impact. It is also found that the users with different parking durations have different levels of sensitivity, resulting that each group receives its own parameter. The optimized price mechanism ensures a lower charge for short-term parking behavior and a higher charge for long-term parking behavior, so that the preferred users can benefit from on-street parking while the non-preferred users are restricted from occupying on-street parking spaces. The price discrimination of on-street parking of Guangzhou city is selected to test the proposed method, and the results show that the price scheme for the high-demand areas is reasonable and the price scheme for the low-demand areas should be improved, which is also proved by the surveys.

The future studies can further consider the research on parking charges in road sections such as night parking and dedicated parking spaces, to make more efficient use of on-street parking spaces. The proposed method can not only be used to optimize on-street parking charges, but also to propose new solutions to other urban parking problems such as parking sharing. In addition, it can also provide a new evaluation index for price discrimination in the situations that have multiple choices.

**Author Contributions:** Conceptualization, J.L.; funding acquisition, J.L.; investigation, S.W. and X.F.; methodology, S.W.; software, S.W.; supervision, J.L.; validation, J.L.; writing—original draft, S.W. and X.F.; writing—review and editing, J.L. All authors have read and agreed to the published version of the manuscript.

**Funding:** This work was supported by the Research and Development Project in Key Areas of Guangdong Province (No. 2019B090913001).

**Institutional Review Board Statement:** Ethical review and approval were waived for this study, due to the human subject involved in this study is about residents' parking behavior. Respondents are recruited on the basis of voluntary and informed consent, and the rights and privacy of the respondents will be protected to the maximum extent possible, with no potential risk to the respondents.

**Informed Consent Statement:** Informed consent was obtained from all subjects involved in the study.

**Data Availability Statement:** The data presented in this study are available on request from the corresponding author. The data are not publicly available due to privacy protection of respondents.

**Conflicts of Interest:** The authors declare no conflict of interest.

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
