# Peer review of "Optimization of On-Street Parking Charges Based on Price Elasticity of the Expected Perceived Parking Cost"

_sustainability, doi:10.3390/su13105735_

Round 1

Reviewer 1 Report

The authors in this paper propose an optimization model of pricing mechanism based on expected perceived parking cost (EPPC) to provide on-street parking spaces to travelers who need them the most and adjust parking distribution. The perceived cost model aims to adjust the charging price within the acceptable pricing range of the user and maximize the influence and guidance of the charging mechanism on consumer behavior. The overall purpose of the mechanism is to implement a lower charge for short-term parking behavior and a higher charge for long-term parking behavior, which can not only allow the consumers to benefit from on-street parking but also restrain the behavior of occupying on-street parking space for a long time. This is likely to improve sustainability. The authors report their experiences of the proposed mechanism on six center districts of Guangzhou city.   This is a well-written paper. Some proofreading is needed to improve the quality of writing and remove typos.   A more detailed literature review and the case for work would improve the reader's interest and confidence.      

Reviewer 2 Report

Dear Authors,

It is an interesting topic of research on regulating the on-street parking charges and have a model that works for both consumers and suppliers. As such, a new index of price elasticity is very good.

My minor comment is to have a simple table of the proposed scheme that is useful for worldwide authors and organisation to use this model in future. For example Table 7 can be redrawn with generalised currency and methods.

Reviewer 3 Report

Try to identify better contribution and avoid this is a technical case about China case. Our proposed approach how can be replicated and used

Literature review try to discuss better and perform idea systematization towards our contribution

Add papers from MDPI to increase ranking - in this topics, several comes like https://www.mdpi.com/1996-1073/12/11/2123

Introduce major findings and at conclusion improve what we win with the current work

Minor corrections

Abstract at end remove one '.'

References needs space, avoid thinks like this  reduced[34], therefore, traffic jams and environmental pollution can be decreased[35], 

should be  reduced [34], therefore, traffic jams and environmental pollution can be decreased [35],
